# Genetically Modified Cellular Therapies for Malignant Gliomas

**DOI:** 10.3390/ijms222312810

**Published:** 2021-11-26

**Authors:** Michael Kilian, Theresa Bunse, Wolfgang Wick, Michael Platten, Lukas Bunse

**Affiliations:** 1DKTK (German Cancer Consortium), Clinical Cooperation Unit (CCU), Neuroimmunology and Brain Tumor Immunology, German Cancer Research Center (DKFZ), 69120 Heidelberg, Germany; m.kilian@dkfz.de (M.K.); t.bunse@dkfz.de (T.B.); m.platten@dkfz.de (M.P.); 2Department of Neurology, Medical Faculty Mannheim, MCTN, University of Heidelberg, 68167 Mannheim, Germany; 3Neurology Clinic, Heidelberg University Hospital, University of Heidelberg, 69120 Heidelberg, Germany; wolfgang.wick@med.uni-heidelberg.de; 4DKTK CCU Neurooncology, DKFZ, 69120 Heidelberg, Germany; 5Immune Monitoring Unit, National Center for Tumor Diseases (NCT), 69120 Heidelberg, Germany; 6Helmholtz-Institute of Translational Oncology (HI-TRON), 55131 Mainz, Germany

**Keywords:** adoptive T cell transfer, glioblastoma, glioma, brain tumor, TCR, CAR

## Abstract

Despite extensive preclinical research on immunotherapeutic approaches, malignant glioma remains a devastating disease of the central nervous system for which standard of care treatment is still confined to resection and radiochemotherapy. For peripheral solid tumors, immune checkpoint inhibition has shown substantial clinical benefit, while promising preclinical results have yet failed to translate into clinical efficacy for brain tumor patients. With the advent of high-throughput sequencing technologies, tumor antigens and corresponding T cell receptors (TCR) and antibodies have been identified, leading to the development of chimeric antigen receptors (CAR), which are comprised of an extracellular antibody part and an intracellular T cell receptor signaling part, to genetically engineer T cells for antigen recognition. Due to efficacy in other tumor entities, a plethora of CARs has been designed and tested for glioma, with promising signs of biological activity. In this review, we describe glioma antigens that have been targeted using CAR T cells preclinically and clinically, review their drawbacks and benefits, and illustrate how the emerging field of transgenic TCR therapy can be used as a potent alternative for cell therapy of glioma overcoming antigenic limitations.

## 1. Introduction

Despite intensive research over the last decades, standard of care (SOC) treatment for malignant gliomas is still restricted to resection and radiochemotherapy. The tremendous clinical effects of immune checkpoint inhibition (ICI) have revolutionized therapy for many cancer entities such as melanoma but have not conferred clinical benefit to brain tumor patients, yet despite promising preclinical results [1,2]. However, none of the phase 3 clinical trials using checkpoint-inhibiting molecules in gliomas met their primary clinical endpoints for patients with newly diagnosed or relapsed glioblastoma (GBM) (Checkmate 143, 498) [3,4,5]. Conversely, more recently, two independent phase 2 trials showed response of neoadjuvant programmed cell death protein 1 (PD-1) therapy in recurrent and operable GBM with response-associated distinct immunogenomic features [6,7,8].

Cellular therapies have become an emerging field in preclinical and clinical cancer research. The first cellular therapies in solid tumors were conducted in 1980 by Rosenberg and colleagues using expanded tumor-infiltrating leukocytes (TIL) and high dose interleukin (IL) 2 [9,10,11]. For brain tumors, TIL therapy in patients with GBM and melanoma brain metastases has been investigated [12,13,14]. However, although promising in some post-hoc analyzed subgroups, the overall outcome of these trials was unsatisfactory even though ex vivo TIL cultures from GBM patients have been shown to exert tumor reactivity [15]. As the usage of endogenous T cells comes with a variety of caveats, such as potential incomplete in vitro reinvigoration of exhausted TIL, limited capacity of TIL to expand in vivo after a strong preceding in vitro stimulation, and potential predominant expansion of bystander T cells, the use of genetically modified T cells could circumvent these obstacles. In recent years, there has been remarkable effort in identifying suitable targets for cellular glioma immunotherapy [16,17]. Chimeric antigen receptor (CAR) T cells have shown tremendous effects in non-solid tumors such as multiple myeloma and leukemia and have recently been approved by the U.S. Food and Drug Administration (FDA) and European Medicines Agency (EMA). For solid tumors, a plethora of early CAR T cell clinical trials has recently been initiated [18]. CARs are designed by using an antibody-derived extracellular recognition domain, a hinging transmembrane domain, and an intracellular T cell receptor (TCR)-derived signaling domain. The antibody-derived variable regions are able to recognize extracellular domains and proteins and bypass major histocompatibility complex (MHC) expression and presentation by tumor cells or professional antigen presenting cells (APC). Alternatively, modified natural ligands of surface receptors may be used as extracellular recognition domains. Modifying the intracellular signaling domain and the addition of co-stimulatory signals has led to the development of second, third, and fourth-generation CARs [19]. In preclinical studies, several CARs against glioma-associated target structures have been developed. In this review, we focus on CAR T cell therapies, highlighting targets for such therapy (Figure 1), we then discuss early phase clinical trials (Table 1), and elaborate on the benefits and drawbacks of CAR T cell therapy, especially in comparison to TCR-engineered T cell therapy. Attention will be turned to the consideration of application routes.

## 2. Glioma Antigens for CAR-Engineered T Cell Therapy

### 2.1. IL13Ra2

Interleukin-13 receptor subunit alpha-2 (IL13Ra2) was the first target in GBM to have been exploited for CAR T cell therapy. As required for tumor-associated targets, IL13Ra2 is highly expressed in a high frequency on tumor cells of GBM patients [20,21,22]. In contrast, healthy tissue, except the testis, express low levels of IL13Ra2, making it an adequate target for targeted therapy. The ubiquitously expressed IL13Ra1 binds IL13 with lower affinity than IL13Ra2, allowing for predominant targeting with modified IL13 variants [23,24,25]. The first CAR targeting IL13Ra2 was developed in 2004 and included a so-called zetakine, which was composed of an extracellular altered IL13 domain tethered to an immunoglobulin transmembrane domain and a CD3-zeta cytoplasmic part [26]. In vitro and in vivo studies in human xenografts showed specific and effective tumor lysis. Second-generation CARs, which include a co-stimulatory 4-1BB domain, outperformed the first-generation CARs while not leading to cytotoxicity in non-obese diabetic (NOD) scid gamma (NSG) mice [27]. Further improvements, such as the addition of an activation-induced expression of IL15 in CAR T cells, led to increased CAR proliferation, cytokine production and persistence [28]. The first in-human clinical trial evaluating IL13Ra2-specific CARs was completed in 2015 and enrolled three patients with recurrent GBM being treated with an IL13-zetakine CAR product [29]. The CAR product was injected directly into the resection cavity in 12 doses over 5 weeks. Overall, the treatment was well tolerated in these patients with only temporary brain inflammation events, indicating promising tolerability of T cell products. Of note, a decrease in tumoral IL13Ra2 expression after CAR therapy was reported in one patient, suggesting therapy-driven antigen loss. On basis of an outstanding casuistic report on a patient with recurrent GBM treated with a 4-1BB-modified CAR product [30], intraventricular IL13Ra2 CAR T cell therapy is currently assessed in a phase 1 clinical trial for ependymoma, GBM, and medulloblastoma (NCT04661384), while intratumoral delivery is being tested in recurrent or refractory malignant glioma (NCT02208362).

### 2.2. Her2

The human epidermal growth factor receptor 2 (Her2) constitutes a GBM-associated antigen, being expressed in approximately 80% of all GBM patients [31,32]. Her2-specific CAR T cells demonstrated preclinical efficacy in several tumor models, were shown to effectively target Her2-positive glioma cells and glioma stem cells, and to lead to regression of GBM xenografts [31,33,34,35,36]. Interestingly, the xenograft study used syngeneic T cells and GBM cells in a patient-derived xenograft (PDX) model, avoiding allogenic immune responses. In another study, Ahmed et al. showed sustained regression of medulloblastoma xenografts in immunodeficient mice using intratumoral adoptive transfer of Her2-specific CAR T cells. As 40% of medulloblastomas overexpress Her2, CAR T cell therapy might represent a promising therapeutic option [37]. Recently, Her2-specific CAR NK cells, derived from the human NK cell line NK-92, have been reported to specifically lyse GBM-derived cell lines and to show in vivo anti-tumor activity in xenografts and immunocompetent mouse models [38]. This NK cell-line-based concept is currently investigated in a phase 1 clinical trial (NCT03383978). Overall, the clinical usage of HER2-specific CARs has been challenged by a case report in 2010 where the administration of a HER2-directed CAR product in a patient with metastatic colon cancer led to a severe and lethal cytokine storm [39]. More recently, subsequent clinical studies reported no severe systemic toxicities [40,41]. In GBM, a phase 1 dose escalation study using HER2-specific CAR T cells derived from virus-specific T cells (VST) showed tolerability with no dose-limiting toxic effect in 17 patients [42]. However, the HER2-VST CARs did not expand in the peripheral blood and clinical efficacy was limited with a median OS of 11.1 months. An interim analysis of a phase 1 clinical trial using locoregional delivery of HER2-specific CAR T cells recently reported no dose-limiting toxicity of multiple CAR T cell infusions and demonstrated highly elevated interferon-induced C-X-C motif chemokine ligand 10 (CXCL10) and CC-chemokine ligand 2 (CCL2) levels in the cerebrospinal fluid (CSF) after CAR T cell infusion [43]. Magnetic resonance (MR) imaging showed vasogenic edema and local intensified contrast enhancement, indicating an inflammatory response called pseudoprogression. The associated clinical trial is composed of two arms, delivering the CAR T cell product into the tumor cavity or intraventricularly, respectively (NCT03500991).

### 2.3. EGFRvIII

The mutated epidermal growth factor receptor variant III (EGFRvIII) results from an amplification of the wildtype EGFR and is expressed in approximately 30% of all GBM patients [44,45,46]. Vaccination or antibody treatment against EGFRvIII has been shown to induce increased survival and long-term immunological memory in various preclinical models [47,48,49], indicating that EGFRvIII can serve as a potent target for cellular therapies. The feasibility of using EGFRvIII-directed CARs has been extensively studied. In a comprehensive study, Morgan et al. screened seven different antibodies for the ability to be used in a CAR T cell product [50]. CAR T cells based on three of these seven antibodies were shown to produce effector cytokines in response to EGFRvIII-expressing glioma cells. In a follow-up study, the authors could show preclinical efficacy of a murine third generation EGFRvIII-directed CAR product in a syngeneic mouse model [51]. Since this study was conducted in immunocompetent mice, the authors reported two important key findings in their preclinical glioma model: first, it was shown that lymphodepletion is needed for a response to systemically injected CAR T cells. Second, CAR T cell therapy was able to induce endogenous long-term immunity against the tumor, even when mice were rechallenged with tumor cells not expressing EGFRvIII. Another study using EGFRvIII CAR T cells showed that overexpression of the mircoRNA miR-17-92, a microRNA that has been reported to enhance T cell survival and interferon (IFN)-γ production and to be downregulated in GBM-infiltrating T cells, increases EGFRvIII CAR T cell function [52]. Interestingly, EGFR806-CAR T cells, recognizing not only EGFRvIII but also full-length gene amplified wildtype EGFR, retained their specificity due to specific steric accessibility of tumor-expressed EGFR [53]. A first in-human study using EGFRvIII-directed CARs was published in 2017 by O’Rourke and colleagues [54]. The CAR design used for this study was selected based on a comprehensive preclinical study [55]. Ten patients were treated with a single peripherally infused dose of the CAR T cell product. Upon recurrence, tissue analysis post CAR T cell transfer revealed that CAR T cells were effectively trafficking to the brain. Reduced EGFRvIII expression in infiltrated regions was rather representing the natural cause of the disease than a sign of immunological escape [56]. However, CAR T cell infiltration led to an increase in T regulatory cell abundance and increased expression of inhibitory molecules such as programmed death-ligand 1 (PD-L1), transforming growth factor ß (TGF-ß), and IL10. More recently, EGFRvIII-specific CAR T cell clinical trials for intracranial tumors, including GBM, have been closed prior to completion due to lack of funding, observed toxicity, shift towards next CAR T cell iteration or combinatorial treatments, or missing objective clinical responses (NCT01454596, NCT02664363, NCT02209376, NCT03283631).

### 2.4. GD2

The disialoganglioside GD2 is frequently overexpressed in neuroblastoma with only restricted expression in healthy tissue, classifying it a tumor-associated antigen. CAR T cell therapy was able to abrogate tumor progression in a xenograft model [57]. CAR T cells against GD2 have also shown tremendous preclinical efficacy in PDX models of H3.3.K27M-mutated midline gliomas [58]. Using second-generation 4-1BB-overexpressing CAR T cells, tumors were almost cleared from different localizations (pons, thalamus, and spinal cord) with a small amount of GD2-negative tumor cells remaining. However, accompanying the strong anti-tumor effect, the authors also observed severe neuroinflammation in immunodeficient mice. GD2 CAR T cell therapy is currently tested in a phase 1 trial in diffuse midline gliomas with retroviral vectors manufactured in the closed CliniMACS Prodigy system (NCT04196413), in combination with constitutive active IL7 receptor (NCT04099797), or in recurrent gliomas (NCT03423992). In the terminated above-mentioned phase 1 clinical trial (NCT04196413) using a retro-viral (14g2a-CD8.BB.z.iCasp9) expressing GD2-chimeric antigen receptor, three of four patients exhibited marked improvement or resolution of neurological deficits as well as radiographic improvements. Moreover, no on-target off-tumor toxicity was observed [59].

### 2.5. Chlorotoxin

Chlorotoxin, a venom-derived peptide, has been described to specifically bind to GBM cells [60,61]. Recently, researchers developed a chlorotoxin-based CAR with the peptide as targeting domain that efficiently targeted tumors with heterogenous expression of GBM-associated antigens such as IL13Rα2, HER2, and EGFR [62]. Of note, the chlorotoxin-based CAR required matrix metalloproteinase 2 (MMP2) expression on the tumor cells for efficient binding. Antitumor capacity was assessed in orthotopic GBM PDX models without severe off-target effects after systemic intravenous or regional intraventricular or intratumoral CAR T cell delivery. A phase 1 clinical trial assessing CAR T cells with a chlorotoxin tumor-targeting domain in MMP2-positive recurrent or progressive GBM is currently recruiting patients (NCT04214392).

### 2.6. EphA2

Ephrin type-A receptor 2 (EphA2) is a receptor tyrosine kinase that binds ephrin-A family ligands and its downstream signaling participates in migration, proliferation, differentiation, and integrin-mediated adhesion [63,64]. It has been described as a glioma-associated antigen with limited expression in healthy tissue with the exception of some epithelial cells [65]. Its overexpression has been reported in several tumor types and linked to decreased overall survival in patients with GBM [66]. Several preclinical studies used EphA2-directed CAR T cells to treat GBM xenografts and showed potent anti-tumor activity against glioma-initiating cells [67,68]. Preclinical locoregional delivery of CAR T cells was validated as effective treatment in medulloblastoma mouse models [69]. However, to date, to our knowledge, clinical studies evaluating EphA2-directed CAR T cells have not yet been initiated.

### 2.7. P32

P32, also known as complement component 1 Q subcomponent-binding protein (C1QBP), has previously been reported to be expressed in tumor cells and tumor-associated endothelial cells [70]. More recently, Rousso-Noori et al. reported P32 to be specifically expressed on murine and human glioma cells [71]. In their study, CAR T cell therapy was able to reduce tumor growth in xenograft and syngeneic mouse models. The authors used a combination of intratumoral and intraventricular injection of the CAR T cell product, leading to sustained infiltration of CAR T cells into the tumor. Conceptualization of clinical trials investigating P32 as CAR T cell target can be expected.

### 2.8. CD133

CD133 is a pentaspan transmembrane glycoprotein reported to be predominantly expressed on cancer, hematopoietic, and neural stem cells [72,73,74]. Several studies have reported CD133 to be involved in tumor initiation and resistance to radio- and chemotherapy [75,76,77]. Upregulation is considered to be prognostically unfavorable. In a side-by-side comparison, while different modalities against CD133 showed efficacy in orthotopic GBM xenografts, CD133-specific CAR-T cells represented the most efficacious [62]. Interestingly, in hematopoietic stem-cell-humanized NOD scid gamma (NSG) mice, intraventricular injection of CD133-specific CAR T cells was effective and did not lead to reduced frequencies of CD34 CD133 double positive hematopoietic cells [73]. As of now, no phase 1 clinical trial using CD133-specific CAR T cells has been initiated.

### 2.9. CSPG4

Chondroitin sulfate proteoglycan 4 (CSPG4) is a type I transmembrane protein that is overexpressed in 67% of GBM [78]. GBM with high CSPG4 expression are considered to be more aggressive than their low expressing counterparts. Specific and cell ratio-dependent killing was observed when CSPG4-specific CAR T cells were co-cultured with CSPG4-expressing primary GBM cell lines or when injected intratumorally in human GBM-bearing nude mice. In these PDX model, no relevant post-treatment antigen loss was observed, which is suggestive for homogeneous target expression in primary GBM cell lines and throughout in vivo growth. To our knowledge, thus far, no clinical trials have been initiated targeting CSPG4 in gliomas.

### 2.10. B7-H3

B7 Homolog 3 (B7-H3) is a type I transmembrane protein that is overexpressed in 76% of GBM [79]. Using established human glioma cell lines such as U87 in nude mice, intratumorally injected B7-H3-specific CAR T cells lead to durable responses independent of applied co-stimulatory domains. Similarly to observations by Nehama et al., Tang et al. reported increased preclinical survival after intravenous injection of B7-H3-specific CAR T cells in U87-bearing NOD SCID mice [80]. More recently, B7-H3 has been successfully co-targeted by B7-H3-CD70 tandem CARs (Tan-CAR) [81]. Compared to single targeting of CD70 or B7-H3, respectively, improved preclinical survival was observed when Tan-CAR T cells were adoptively transferred in non-glioma PDX. Interestingly, B7-H3 co-expression was also observed in glioma. Currently, three clinical trials investigating B7-H3-specific CAR T therapy in recurrent or refractory GBM or various other central nervous system (CNS) tumors are recruiting patients (Table 1).

## 3. Preclinical Strategies to Overcome Off-Target and On-Target Off-Tumor Side Effects

Beyond potential genotoxicity (due to the risk of oncogenic insertional mutagenesis by viral expression vectors), immunogenicity (of CAR epitopes), and off-target side effects, (i.e., interaction of CARs with Fc receptors expressed on myeloid cells inducing antigen-independent T cell activation), severe on-target off-tumor side effects can occur, because CAR T cells usually target tumor-associated rather than tumor-specific antigens. For clinical translation in brain tumors, it is essential to carefully assess potential on-target off-tumor side effects that can lead to severe neuroinflammation and neurotoxicity. Recently, in an observational study, Parker and colleagues identified CD19-expressing brain mural cells as a potential on-target off-site in CD19-specific CAR T cell therapy [82]. Interestingly, on-target off-site-relevant CD19 expression was found in human brain mural cells but not in murine systems, highlighting the limitations of studies on off- and on-target side effects in mouse model systems. One option to overcome on-target off-tumor side effects is the insertion of additional targetable epitopes in CAR constructs themselves [83]. In a preclinical study using an acute myeloid leukemia xenograft mouse model, Bonifant et al. introduced a CD20 domain into their CAR construct, enabling elimination of CAR T cells via rituximab, a CD20-specific therapeutic antibody, if required [84]. To our knowledge, such strategy thus far has not been pursued in the context of preclinical brain tumors. The design of dual CAR constructs could potentially prevent CAR activation distant from the tumor site. By using a synNotch receptor in a bilateral K562-bearing xenograft model, Roybal and colleagues could show specific CAR expression on T cells only after delivery of a first expression-inducing signal [85]. After antigenic ligand encounter, synNotch receptor activation leads to cleavage of a transcription factor within its intracellular signaling domain, which enables expression of the CAR specific for a second antigen. Recently, synNotch CAR T cells that are activated by local and tumor-specific antigen encounters have shown to lead to a specific and locally restricted anti-tumor response [86]. The authors used the CNS- or tumor-specific antigens myelin oligodendrocyte glycoprotein (MOG) or EGFRvIII, respectively, as activating signal for the expression of anti-EphA2 or anti-IL13Ra2 CARs. Using this system, the authors showed tumor microenvironment-restricted CAR expression, T cell priming, and effective killing of tumor cells in PDX models.

## 4. Overcoming Resistance to CAR T Cell Therapy

Shared features of resistance to CAR T cell therapy in non-solid tumors and concepts to overcome these obstacles have comprehensively been discussed elsewhere [87,88,89,90]. In particular, gliomas are signified by a profoundly immunosuppressive microenvironment. Therefore, CAR T cell combinatorial therapies with immunomodulatory agents seem obligatory and have been comprehensively discussed elsewhere [91]. In brief, Agliardi et al. reported that the intratumoral application of IL12 increases response to EGFRvIII CAR T cell therapy by boosting cytotoxicity and remodeling the tumor microenvironment to a more proinflammatory state [92]. Anti-GD2 CARs have been modified with an IL15 cytokine domain and CAR-expressing T cells showed prolonged anti-tumor activity upon repetitive antigen exposure [93], overcoming exhaustion. In GBM xenografts, temozolomide-induced lymphodepletion enhanced CAR T cell expansion and persistence and prolonged survival of tumor-bearing mice [94]. Antigen loss is considered a frequent tumor escape mechanism when only single tumor antigens are targeted. Several reports have shown improved CAR product efficacy and reduced resistance to CAR therapy when multiple antigens were targeted in one therapeutic approach [69,95,96,97]. However, directing CAR T cells towards multiple targets also increases the risk for off-target effects, and clinical trials targeting multiple antigens have not yet been initiated. Loss of antigen as underlying mechanism of CAR T cell resistance is not strictly linked to mutational events; frequently, downregulation of tumor antigens is mediated by epigenetic silencing. Thus, it is tempting to modulate epigenetic silencing of CAR T cell targets using epigenetic drugs. In several entities, blocking DNA methyltransferases in combination with CAR T cell therapy showed signs of improved efficacy in preclinical models [98,99,100,101]. In medulloblastoma xenograft models, the demethylating agent azacytidine led to increased target expression and prolonged survival upon CAR T cell therapy [69].

## 5. Application Routes for Genetically Modified T Cells in Preclinical Models

First preclinical studies for other tumor entities used a systemic intravenous injection of CAR T cell products, resulting in potent anti-tumor effects. Using Her2 CAR T cells in an intracranial breast cancer metastasis model, Priceman and colleagues reported improved tumor eradication rates after intratumoral or intraventricular adoptive transfer [102], and Brown et al. showed no therapeutic effect at all for systemic intravenous injection [27]. Recently, two studies from Theruvath and colleagues and Donovan and colleagues reported enhanced anti-tumor efficacy of locally transferred CAR T cells in several xenograft models of brain tumors. Donovan et al. showed increased survival of intraventricularly injected mice compared to intravenously injected mice in medulloblastoma and ependymoma models using CAR T cells against EPHA2, HER2, and IL13a2 [69]. Theruvath et al. used previously described B7-H3-targeting CAR T cells to show improved survival and reduced systemic inflammatory cytokine levels upon locoregional delivery [103,104]. Particularly preclinically, the literature implies a superior performance of locally administered CAR T cells over systemic application. Nonetheless, all studies were performed in fully immunodeficient mice, while effective trafficking of intravenously injected genetically manipulated T cells to the parenchyma requires cytokine gradients that will not be established in immunodeficient mice. Therefore, more investigations of how transferred T cells efficiently migrate to the brain are required in immunocompetent mice in order to draw definitive conclusions. Likewise, clinical response to CAR T cell therapy is associated with an induction of endogenous tumor-specific T cell responses and microenvironmental reprogramming [105]. Immunocompetent model systems displaying syngeneic MHC-proficient microenvironments represent a pre-requisite to assess therapeutic efficacy, particularly for T cell receptor-engineered T cell therapy targeting T helper cell epitopes, as discussed in the following section.

## 6. T Cell Receptor-Engineered T Cell Therapy

In contrast to CAR T cells, which are able to target only extracellular, mostly non-tumor-specific targets, TCR-transgenic T cells provide the possibility to target intracellular antigens that are presented on MHC class I and II. These targets can be glioma-associated antigens that have been reported to be overexpressed in brain tumors such as SART1 or MAGE1 [106,107,108]. Initial clinical trials in several tumor entities have reported promising results with TCR-engineered T cell therapy [109,110,111]. Most reports to date are based on tumor-associated antigens that are shared across a broad patient population [112,113,114]. Although, the use of tumor-associated antigens led to severe off-target and on-target off-tumor side effects in many early trials [82,113,115], however, the true strength in using transgenic TCR therapy lies in the ability to target tumor-specific mutations, without targeting heathy tissue. Yet, GBMs harbor only 30 to 50 mutations, leading to a limited repertoire of potential targets [116,117,118]. Recently, two clinical trials could show that vaccination against actively personalized or warehouse patient-specific neoepitopes can trigger T cell-driven immune responses [119,120]. Tumor-specific mutations are usually patient-individual mutations, leading to a prolonged time for target identification and therapy manufacturing. This could be circumvented by using frequently mutated targets. We and others have shown T cell immunogenicity for several common mutations in brain tumors, as reported for isocitrate dehydrogenase I (IDH1) mutation R132H, histone H3.3 mutation K27M, and more recently capicua transcriptional repressor (CIC) R215W/Q [121,122,123,124,125,126,127]. As these targets are also known to be driver mutations, the risk of antigen loss is reduced [128,129,130,131,132]. However, specific driver mutations such as mutant IDH1 have been reported to have immunosuppressive capacities themselves by, for example, tumor-genotype dependent education of myeloid cells [133,134,135] or suppression of antitumor T cell responses [123,136,137] in the tumor microenvironment and in peripheral immune compartments [138]. These observations have important implications for the future development of locoregional or systemic cellular therapies for IDH mutant gliomas.

In summary, an ideal target for T cellular therapy would be tumor-specific, shared by a significant cohort of patients, and a driver mutation to reduce the risk of immune escape by antigen loss. The first glioma-specific TCR that has been reported targets the K27M mutation in histone H3.3 [139]. Despite being able to detect vaccine-induced T cell clones in the periphery of diffuse midline gliomas (DIPG) patients, the authors could retrieve an H3.3K27M-reactive TCR sequence from peptide-pulsed healthy donor peripheral mononuclear cells. In in vitro and in vivo assays against an H3.3K27M-transfected human GBM cell line, H3.3K27M-specific TCR-transduced primary T cells specifically lysed tumor cells and inhibited tumor progression. More recently, we described a TCR targeting the IDH1 R132H mutation [122]. Using single cell transcriptomic and TCR sequencing, we were able to retrieve an IDH1 R132H-specific TCR from a post peptide vaccine inflamed CNS lesion. This example demonstrates that, with the emerging accessibility of single cell sequencing, identification of tumor-reactive and patient-specific TCRs will be increasingly achievable, hence exploitable for therapeutic approaches [140,141].

## 7. Perspective

Main hurdles for efficacious cell therapies for brain tumors will remain sufficient infiltration, persistence, and resilience of genetically modified T cells into or within the hostile brain tumor microenvironment. More recent studies suggest an IDH mutation status-associated reduced antigen presentation capacity and particularly profound exclusion of T cells within or from the IDH mutant tumor microenvironment. Despite proneural to mesenchymal transitions, mesenchymal subtype glioblastomas are considered an immunologically more active glioblastoma subtype [142]. These glioma entity- or glioblastoma subtype-specificities are suggestive for potentially required appropriate adjustments of cellular therapeutics. 

Current concepts to overcome the main hurdles include the utilization of alternative application routes and combinatorial treatments in preclinical and in early clinical trials. Some evidence of relevant peripheral antigen presentation in brain tumors exists [143,144]. As of now, however, it remains unclear if antigen presentation has an impact on efficacy of local versus systemically applied TCR- or CAR-engineered T cells in gliomas. If systemic antigen presentation proves to be relevant, synthetic vaccine-based in vivo CAR T cell boosting concepts, which have already been designed [145], will gain importance. Some trials are even terminated prior to completion to pursue combination therapies with i.e., checkpoint inhibition instead. CAR- and TCR-engineered T cell therapy have both unique advantages and disadvantages, and combinations of both cellular concepts should be explored preclinically in the near future. At the same time, inducible cellular systems are in preclinical development to reduce on-target off-tumor side effects. Whereas CAR T cells target predominantly tumor-associated cell surface antigens, TCR-engineered T cell therapy will strictly depend on intratumoral MHC class I or class II expression and antigen presentation by glioma cells and associated professional antigen-presenting cells, respectively (Figure 1). This prerequisite should be considered when conceptualizing clinical trials investigating TCR-engineered T cell therapy. In this regard, for biomarker assessment, preceding biopsies or resected tumor tissues should be representative for the current tumor disease at time of treatment start.

## 8. Conclusions

In this review we discuss emerging targets and recent preclinical and clinical developments in the field of CAR- and TCR-engineered T cell therapy for malignant gliomas. In comparison to other immunotherapeutic modalities for gliomas, such as dendritic cell vaccination, peptide vaccination, or TIL therapy, CAR- and TCR-engineered T cell therapy offers several advantages: (i) independence of the patient’s immune system to mount meaningful T cell responses, which is, in general, limited in vaccination approaches; (ii) immune receptor engineered T cells target known and defined antigens; (iii) T cells can be additionally genetically modified ex vivo to enhance T cell responses; (iv) in contrast to TIL therapy, the re-infusion of an engineered T cell product allows the exact enumeration of truly tumor-reactive T cells; and (v) defined TCR as well as CAR sequences or epitopes facilitate the analysis of T cell fate and dynamics during monitoring. Up to now, however, these advantages have been inseparable of great financial and regulatory challenges in the processes of manufacturing and engineering cellular products. At the same time, CAR- and TCR-engineered T cell therapy bears the potential—as in other tumor entities–to cure gliomas.

## Figures and Tables

**Figure 1 ijms-22-12810-f001:**
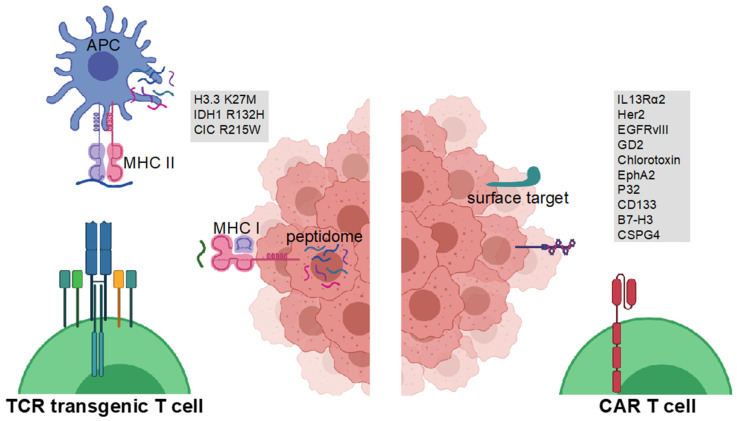
Glioma antigens for CAR- and TCR-engineered T cell therapy. TCR-engineered T cells target MHC class I-bound short peptides or MHC class II-bound long glioma-specific peptides on glioma cells or glioma-associated myeloid professional antigen-presenting cells (APC), respectively (left). CAR T cells target cell surface proteins on glioma cells (right). Figure created with BioRender.com.

**Table 1 ijms-22-12810-t001:** Clinical trials investigating genetically modified cellular therapies in brain tumors.

Clinical Trial	Entity	Target	Start	Phase	Combination
NCT04196413	Diffuse intrinsic pontine gliomas (DIPG) + Spinal diffuse midline glioma (DMG)	GD2	Dec 19	1	Fludara, Cyclo
NCT04003649	Recurrent or refractory GBM	IL13Ra2 + Ipi	Jul 19	1	Nivo + Ipi
NCT02442297	HER2-positive CNS tumors	Her2	May 15	1	
NCT04510051	Recurrent or refractory brain tumors in children	IL13Ra2	Aug 20	1	Fludara, Cyclo
NCT04099797	GD2-positive brain tumors	GD2	Sep 19	1	Fludara, Cyclo
NCT04661384	Leptomeningeal GBM, ependymoma, or medulloblastoma	IL13a2	Dec 20	1	
NCT04185038	DIPG/DMG and recurrent or refractory pediatric CNS tumors	B7-H3	Dec 29	1	
NCT03638167	EGFR-positive recurrent or refractory pediatric CNS tumors	EGFR806	Aug 18	1	
NCT03500991	HER2-positive recurrent or refractory pediatric CNS tumors	HER2	Apr 18	1	
NCT03696030	Recurrent brain or leptomeningeal metastases	HER2	Oct 18	1	
NCT04903795 *	Grade 4 malignant glioma	EGFRvIII	May 21	1	autologous activated T cells
NCT04385173	Recurrent and refractory GBM	B7-H3	May 20	1	TMZ
NCT04077866	Recurrent or refractory GBM	B7-H3	Sep 19	1/2	TMZ
NCT04214392	MPP2-positive recurrent or progressive GBMa	Chlorotoxin-derived CAR	Jan 20	1	
NCT04045847	Recurrent malignant glioma	CD147	Aug 19	1	
NCT03389230	Recurrent or refractory grade 3–4 glioma	Her2	Jan 18	1	
NCT02208362	Recurrent or refractory malignant glioma	IL13Ra2	Aug 14	1	
NCT02575261 #	EphA2-positive malignant glioma	EphA2	Oct 15	1/2	
NCT03726515 $	Newly diagnosed MGMT-unmethylated GBM	EGFRvIII	Oct 18	1	Pembro
NCT03170141 ~	GBM	several	May 17	1	Fludara, Cyclo
NCT03423992	Recurrent malignant gliomas	several	Feb 18	1	
NCT03383978	Recurrent HER2-positive GBM	HER2	Dec 17	1	

* Not yet recruiting; # withdrawn; $ completed; ~ enrolling by invitation; Cyclo, Cyclophosphamide; DIPG, diffuse intrinsic pontine gliomas; DMG, diffuse midline glioma; Fludara, Fludarabine; Ipi, Ipilimumab; Nivo, Nivolumab; Pembro, Pembrolizumab; TMZ, temozolomide.

## Data Availability

Not applicable.

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
