# Peer review of "Genetically Modified Cellular Therapies for Malignant Gliomas"

_ijms, 2021, doi:10.3390/ijms222312810_

Round 1

Reviewer 1 Report

In respect to the  recent extensive reviews available on this topic the current manuscript  fails to spark the readers' interest. The main attention of this review is supposedly focussed on the administration route, a topic that legitimately merits attention. However in the current draft this topic consist only in a small paragraph (section 5).

 I agree with the authors that this is an important point that need to be addressed to optimize both efficacy and safety of this approaches.Thus I suggest the author to refocus the topic of their review to increase the novelty and the interest for the readers. The authors could described the anatomical specificity of the brain and possible administration routes while discussing for each route the advantage and limitation.   They should also discuss the mechanism of CAR T cell migration into the brain parenchyma toward the tumor site and discuss  how this can translate in optimized strategy aimed at increasing the tumor trafficking. Experimental models to study this aspect should be provide also.  I also suggest the author to refocus either on adult glioblastoma or on pediatric gliomas which are two different situation with both their specific challenge. I found this point somehow misleading in the current manuscript.

In the conclusion section, the authors should discuss the current limitations in respect to other immunotherapeutic strategies currently under investigation and add a Perspectives Section   

I found the other topics discussed in the other sections redundant in respect with the current recent published review on this topic.    

Other points:

In table 1, the clinical trial should be relevant to glioblastoma. The context in pediatric oncology is completely different and I found it misleading.

In section 2, the list of antigens is incomplete. 

Section 3 need to be re-written and focused on data obtained in small animal and in clinical trial.

Section 4 this topic has been extensively reviewed elsewhere and present little interest for the reader.

Section 6 arrives out of the blue. What's the link with the administration route ?

A Perspective Section should be added before the conclusion.

Author Response

In respect to the recent extensive reviews available on this topic the current manuscript fails to spark the readers' interest. The main attention of this review is supposedly focussed on the administration route, a topic that legitimately merits attention. However in the current draft this topic consist only in a small paragraph (section 5).

 I agree with the authors that this is an important point that need to be addressed to optimize both efficacy and safety of this approaches. Thus I suggest the author to refocus the topic of their review to increase the novelty and the interest for the readers. The authors could described the anatomical specificity of the brain and possible administration routes while discussing for each route the advantage and limitation. They should also discuss the mechanism of CAR T cell migration into the brain parenchyma toward the tumor site and discuss how this can translate in optimized strategy aimed at increasing the tumor trafficking. Experimental models to study this aspect should be provide also. I also suggest the author to refocus either on adult glioblastoma or on pediatric gliomas which are two different situation with both their specific challenge. I found this point somehow misleading in the current manuscript.

We thank the referee for the feedback on our invited review on molecularly engineered cellular therapies for malignant gliomas. In our review, we aimed at balancing comprehensiveness in CAR and TCR T cell therapy and at the same time highlight unique features and concepts in cellular therapies for gliomas not previously discussed in other reviews. With the fruitful input from both referees, we believe that our review has been substantially improved.

In the conclusion section, the authors should discuss the current limitations in respect to other immunotherapeutic strategies currently under investigation and add a Perspectives Section.

In a conclusion and perspective section, we now discuss unique features of potential glioma-entity and subtype-specific requirements for cellular therapies (see also suggestions by referee 2) and discuss advantages and challenges of cellular therapies compared to other immunotherapeutic modalities such as vaccination or TIL therapy in malignant gliomas.

I found the other topics discussed in the other sections redundant in respect with the current recent published review on this topic.

We agree with the reviewer, that the topic is currently of broad interest to the field and therefore intensively discussed in recent reviews, however, to maintain comprehensiveness and to enable the readers to understand the unique aspects of this review, i.e., TCR T therapy, microenvironmental specificities, unique features of CAR vs TCR therapy, and application routes we would suggest to keep the major frame of this review.

Other points:

In table 1, the clinical trial should be relevant to glioblastoma. The context in pediatric oncology is completely different and I found it misleading.

According to the reviewer´s suggestion, we changed the title of table 1, as studies were indeed not restricted to GBM. From an immunological perspective, there are shared obstacles and opportunities in pediatric and adult neuro-oncology applying cellular therapies. Having carefully checked age inclusion criteria in all clinical trials listed in Table 1, in many trials it is difficult to distinguish between adult and pediatric patient cohorts. Some CAR T targets are present in both, pediatric and adult patient populations. Although we agree with the reviewer that clinical differences are important, we would suggest to keep the clinical trials listed.

In section 2, the list of antigens is incomplete.

According to the reviewer´s suggestion, we added further antigens to section 2.

Section 3 need to be re-written and focused on data obtained in small animal and in clinical trial.

We now clearly indicate that, at current stage, these are strategies which have demonstrated efficacy in preclinical models. We rephrased this section accordingly.

Section 4 this topic has been extensively reviewed elsewhere and present little interest for the reader.

We now refer to comprehensive reviews on this particular topic and kept this section short.

Section 6 arrives out of the blue. What's the link with the administration route ?

Current studies investigating differential applications routes are conceptualized in immunodeficient mouse model systems. Endogenous reprogramming of the adaptive and innate immune system has implications for effector cell recruitment to brain tumors. Therefore, before drawing definitive conclusions, additional studies using immunocompetent models are required to systematically compare application routes. The pre-requisite of using immunocompetent model systems is now used to link sections 5 and 6.

A Perspective Section should be added before the conclusion.

We now include a perspective and conclusion section.

Reviewer 2 Report

It has been a pleasure reading this manuscript titled "Genetically modified cellular therapies for malignant gliomas" prepared by Kilian et al., the author covered current research advances in the immune cell therapy for malignant gliomas. They covered several frequently used targets such as IL13Ra2, Her2, and EGFRvIII, etc. I think the content well covered this field, which could be very helpful for the audience. The only thing I hope they could expand the discussion is the application of CAR T therapy in different glioma subtypes. Specifically, glioma with IDH mutation exhibits a distinctive tumor microenvironment and immunological profile.

Author Response

It has been a pleasure reading this manuscript titled "Genetically modified cellular therapies for malignant gliomas" prepared by Kilian et al., the author covered current research advances in the immune cell therapy for malignant gliomas. They covered several frequently used targets such as IL13Ra2, Her2, and EGFRvIII, etc. I think the content well covered this field, which could be very helpful for the audience. The only thing I hope they could expand the discussion is the application of CAR T therapy in different glioma subtypes. Specifically, glioma with IDH mutation exhibits a distinctive tumor microenvironment and immunological profile.

We thank the referee for this very positive and constructive feedback. We now include important references and some comments on unique challenges and considerations regarding the mutant IDH microenvironment but also GBM subtypes in sections 6 and 7.

Round 2

Reviewer 1 Report

Most of the concerns have been taken care by the authors. I found the current manuscript version acceptable for publication in IJMS.